# Enhancing Rumen Fermentation Characteristic and Methane Mitigation Using Phytonutrient Pellet in Beef Cattle

**Thiwakorn Ampapon [1]**, **Theerachai Haitook [2] and Metha Wanapat [2,*]**

1   Department of Animal Science, Faculty of Agriculture and Technology, Rajamangala University of Technology, Isan Surin Campus, Surin 32000, Thailand; ouiyaxx@hotmail.com
2   Tropical Feed Resources Research and Development Center (TROFREC), Department of Animal Science, Faculty of Agriculture, Khon Kaen University, Khon Kaen 40002, Thailand; theerachai.anisci@gmail.com
*   Correspondence: metha@kku.ac.th; Tel.: +66-4320-2362

**Abstract:** The objective of this experiment was to assess the effects of chaya (*Cnidoscolus aconitifolius*) leaf and rambutan (*Nephelium lappaceum* L.) fruit peel pellet (CRP) supplementation on rumen fermentation, feed intake, nutrient digestibility, and nitrogen balance in crossbred beef cattle. Four beef cattle crossbred bulls (Brahman 75% × Thai native 25%), with 250 ± 15 kg of liveweight at 18 months old, were randomly selected to receive four dietary treatment groups: no supplementation (control) and the supplementation of CRP at 2, 4 and 6% of dry matter intake (DMI) by using a 4 × 4 Latin square design. The cattle were fed a concentrate at 2 kg/day, and rice straw was offered ad libitum. The results showed that rumen pH at 4 h and average post-feeding values were in the normal range, though they were slightly reduced with CRP supplementation at 4 and 6% DMI ($p < 0.05$). Rumen temperature, ammonia nitrogen level, and total volatile fatty acid (VFA), acetate ($C_2$), and butyrate ($C_4$) production were similar among treatments. Nevertheless, propionate ($C_3$) concentration was significantly increased ($p < 0.05$) in the CRP groups at 2 and 4% DMI. In addition, the $C_2/C_3$ ratio and $CH_4$ production were significantly reduced in the CRP groups, especially at 2 and 4% DMI. Rice straw intake and total feed intake in terms of %BW were significantly higher in the groups with CRP at 2 and 4% DMI ($p < 0.05$). The apparent nutrient digestibilities were not affected by the treatments. Nitrogen intake, nitrogen absorption, and nitrogen retention were significantly enhanced by the CRP supplementation ($p < 0.05$). Moreover, feces and total nitrogen excretion were not different among treatments ($p > 0.05$). Ultimately, the supplementation of CRP at 2 and 4% DMI significantly improved the $C_3$ concentration, reduced the $C_2/C_3$ ratio, mitigated methane production, and enhanced feed intake and nitrogen utilization efficiency. Therefore, CRP supplementation shows promise as a rumen dietary enhancer.

**Keywords:** phytonutrient; methane mitigation; rumen enhancer; nutrient digestibility; ruminant

## 1. Introduction

Local feed resources are important contributions to livestock production efficiency. There are a number of factors that can enhance rumen fermentation, such as the sources of protein and carbohydrate in different seasons [1]. Traditionally, fodder shrubs and trees are used as feed-additive supplements to supply high levels of protein, as well as carbohydrates, vitamins, and minerals for improving ruminant production [2]. Chaya (*Cnidoscolus aconitifolius*) is a tropical fodder shrub originating from Mexico that can grow well in tropical and sub-tropical countries with yield of about 5 to 8 tons/ha. Chaya is a plant of importance; it has historically served as human food and medicine, as well as a source of protein for animal feeding [3]. Chaya leaves contain a high level of protein (29–31%), essential amino acids, and minerals, as well as flavonoids, tannins, and saponins [4,5]. The supplementation of chaya leaves was shown to improve nutrient digestibility, rumen fermentation, and ruminant production efficiency in in vitro and in vivo study (cattle) [5,6].

Global warming has been the subject of concern from all sectors in which enteric fermentation has contributed by rumen methane production by ruminants [7,8]. There have been many attempts to mitigated the impact of rumen methane emission. Rambutan (*Nephelium lappaceum* L.) peel is an agricultural by-product from the fruit industries of tropical countries. Rambutan fruit peels contain a high concentration of phytonutrients (PTN) including condensed tannin (CT) and saponins (SP), which have impacts on rumen methane suppression [9]. Furthermore, Patra and Saxena [10] stated that the supplementation of PTN could significantly reduce the protozoal population and inhibit methanogenesis in the rumen. Cherdthong et al. [11] confirmed that the use of tropical plants containing CT and SP can increase the concentration of rumen $C_3$ and reduce protozoal population and methanogens, hence alleviating methane production in ruminants. Moreover, CT provides beneficial effects in mitigating the degradation of proteins (bypass proteins) in the rumen and rapidly flowing into the lower gut for later digestion. Thus, feed utilization and livestock production efficiency can be enhanced. A combination of chaya leaves and rambutan peels could enrich the pellet with essential amino acids and phytonutrients from both sources. Nevertheless, no research work regarding the influence of chaya leaves combined with rambutan peels on rumen fermentation characteristics and nutrient digestibility in ruminants has been reported.

Thus, the objective of this experiment was to determine the effect of chaya leaf and rambutan peel pellet (CRP) supplementation on rumen fermentation, feed intake, and nutrient digestibility in crossbred beef cattle.

## 2. Materials and Methods

### 2.1. Feeds, Animals, and Experimental Design

Chaya leaves were collected from shrubs with 60 days of regrowth, then chopped into 1–3 mm pieces, sun-dried, and ground to pass a 1 mm screen using a Cycletech mill (Taeater, Sweden). Fresh rambutan peels were collected from the Malee Fruit Canning Group Public Company Limited, Nakhon Pathom province, Thailand. Peels were cleaned and sun-dried to obtain at least 90% of dry matter and ground to pass through a 1 mm screen mill. CRP pellets consisted of chaya leaf power (70%), rambutan peel powder (20%), cassava powder (9%), and molasses (1%). All ingredients were mixed well and then pelleted using a Ryuzo Kun pelleting machine under hydraulic pressure and a temperature of about 60 °C for a duration of 3–5 min per batch. Then, the pellets were air-dried to obtain at least 88% of dry matter. The concentrate supplement was mixed by using locally available feed ingredients, and rice straw was a roughage source. Feed compositions are reported in Table 1.

**Table 1.** Feed ingredients and chemical composition of the experimental diets.

| Item | Concentrate | Rice Straw | CRP |
|---|---|---|---|
| Ingredients, % air dry basis | | | |
| Cassava powder | 28.0 | - | 9.0 |
| Corn meal | 29.0 | - | - |
| Rice bran | 13.0 | - | - |
| Palm kernel meal | 15.0 | - | - |
| Soybean meal | 8.0 | - | - |
| Molasses | 2.0 | - | 1.0 |
| Urea | 2.0 | - | - |
| Di-calcium phosphate | 1.0 | - | - |
| Mineral mixture | 1.0 | - | - |
| Sulfur | 1.0 | - | - |
| Chaya powder | - | - | 70.0 |
| Rambutan peel powder | - | - | 20.0 |
| Total | 100.0 | - | 100.0 |

**Table 1.** *Cont.*

| Item | Concentrate | Rice Straw | CRP |
|---|---|---|---|
| Chemical composition | | | |
| Dry matter (DM), % | 90.9 | 93.4 | 89.8 |
| | | % dry matter | |
| Organic matter (OM) | 95.0 | 89.0 | 93.1 |
| Crude protein (CP) | 16.0 | 3.1 | 20.1 |
| Neutral detergent fiber (NDF) | 25.3 | 80.1 | 35.3 |
| Acid detergent fiber (ADF) | 12.1 | 49.3 | 17.5 |
| Total digestible nutrients (TDN) * | 75.0 | - | - |
| Crude ash (CA) | 4.5 | 11.0 | 6.9 |
| Ether extract (EE) | 2.7 | 0.9 | 1.5 |
| Condensed tannins (CT) | - | - | 5.1 |
| Saponin (SP) | - | - | 4.8 |
| Mineral, mg/kg | | | |
| Potassium (K) | - | - | 0.48 |
| Calcium (Ca) | - | - | 3.30 |
| Magnesium (Mg) | - | - | 1.65 |
| Phosphorus (P) | - | - | 0.91 |
| Sodium (Na) | - | - | 0.11 |
| Amino acid profiles, mg/g | | | |
| Aspartic acid | | | 1.092 |
| Glutamic acid | | | 2.604 |
| Serine | | | 1.533 |
| Asparagine | | | 4.732 |
| Threonine | | | 1.995 |
| Glutamine | | | 1.232 |
| Tyrosine | | | 15.785 |
| Glycine | | | 0.469 |
| Proline | | | 5.362 |
| Alanine | | | 8.435 |
| Methionine | | | 0.063 |
| Valine | | | 7.616 |
| Phenylalanine | | | 20.020 |
| Isoleucine | | | 12.187 |
| Leucine | | | 14.238 |
| Tryptophan | | | 4.879 |
| Cysteine | | | 0.035 |
| Histidine | | | 0.518 |
| Lysine | | | 1.155 |
| Arginine | | | 4.655 |
| Total | | | 108.605 |

CRP = Chaya powder plus rambutan peel powder pellet; * Calculated value.

Four beef cattle crossbred bulls (Brahman 75% × Thai native 25%), with a liveweight of 250 ± 15 kg at 18 months old, were randomly selected to receive four dietary treatments: no supplementation (control) and chaya leaf and rambutan peel pellet (CRP) supplementation at 2, 4 and 6% of dry matter intake (DMI) (T2, T3, and T4, respectively). Feed composition is reported in Table 2. The cattle were fed a concentrate at 2 kg/day mixed in with treatments, and rice straw was fed *ad libitum*. Four periods of a 4 × 4 Latin square design were employed for 21 d of each period.

**Table 2.** Composition of concentrate plus chaya powder plus rambutan peel powder pellet (CRP).

| Item | Concentrate (2 kg) + CRP (% of Dry Matter Intake) | | | |
|---|---|---|---|---|
| | 0 | 2 | 4 | 6 |
| Chemical composition | | | | |
| Dry matter, % | 95.0 | 94.9 | 94.8 | 94.8 |
| Organic matter | 95.5 | 95.4 | 95.3 | 95.2 |
| Crude protein | 16.0 | 16.1 | 16.3 | 16.5 |
| Neutral detergent fiber | 25.3 | 25.8 | 26.2 | 26.6 |
| Acid detergent fiber | 12.1 | 12.4 | 12.6 | 12.8 |
| Crude ash | 4.5 | 4.6 | 4.7 | 4.8 |
| Ether extract | 2.7 | 2.6 | 2.6 | 2.5 |
| Condensed tannins (CT) | 0 | 0.2 | 0.5 | 0.7 |
| Saponin (SP) | 0 | 0.2 | 0.4 | 0.6 |

*2.2. Data Collection and Samples Analysis*

Feed samples were collected during the first two weeks, and feed intake on a dry matter basis was measured daily. Animals were set on metabolism crates to collect feces and urine for 7 consecutive days at the end of the 21 day period. On the 21st day of each period, about 100 mL of rumen fluid was collected from the rumen at 0 and 4 h post-feeding with a stomach tube connected to a vacuum pump. The rumen fluid was immediately measured for rumen pH and temperature using a pH meter (HI 8424 microcomputer, HANNA instrument, Singapore); then, the rumen fluid samples were filtered, and 45 mL of each sample was collected and mixed with 5 mL of an $H_2SO_4$ solution (1 M) to stop microbial activity for further volatile fatty acid (VFA) and ammonia nitrogen ($NH_3$-N) analyses. The rumen fluid samples were centrifuged at $1600\times g$ for 15 min and stored at –20 °C for further analyses of total VFA (TVFA), acetate ($C_2$), propionate ($C_3$) and butyrate ($C_4$) using high-performance liquid chromatography (HPLC) [12] and for the calculation of the C2/C3 ratio. On the day of rumen fluid sampling, blood samples (approximately 10 mL) were collected from the jugular veins into tubes and mixed with EDTA, centrifuged at $500\times g$ for 10 min at 4 °C, and stored at –20 °C until used for blood urea nitrogen (BUN) analysis, as already described by Crocker [13]. Feeds, urine, and feces were analyzed for chemical composition and used for the estimation of digestibility and nitrogen balance parameters [14,15]. Nutrient digestibilities were calculated using acid-insoluble ash (AIA) as an internal indicator [16]. The protocol of this experiment was previously described by Wanapat et al. [17]. The ruminal $CH_4$ production was estimated using VFA proportions as follows [18]:

$$CH_4 \text{ production} = 0.45(C_2) - 0.275(C_3) + 0.4(C_4) \tag{1}$$

*2.3. Statistical Analyses*

Data were analyzed according to a $4 \times 4$ Latin square design using the GLM procedure of SAS [19]. The main factors were beef cattle, period, and CRP level supplementation. Treatment means were analyzed by polynomial comparison (linear, quadratic, and cubic). Differences among means with $p < 0.05$ are stated as statistical differences.

**3. Results**

*3.1. Feeds and Chemical Composition*

The concentrate was formulated using feed ingredients from locally available sources and comprised 95.0% OM, 16.0% CP, 75.0% TDN, 25.3% NDF, and 12.1% ADF; the rice straw comprised 89.0% OM, 3.1% CP, 80.1% NDF, and 49.3% ADF; the chaya leaf and rambutan peel pellet (CRP) comprised 93.1% OM, 20.1% CP, 35.3% NDF, 17.5% ADF, 5.1% CT, 4.8% SP, 0.48% K, 3.30% Ca, 1.65% Mg 0.91% P, and 0.11% Na. In addition, CRP contained tyrosine, phenylalanine, isoleucine, leucine, alanine and other amino acids at the levels shown in Table 1. The concentrate and CRP compositions are presented in Table 2.

### 3.2. Rumen Fermentation and Methane Estimation

The rumen pH at 4 h and average post-feeding values were linearly reduced following the supplementation of CRP at 4 and 6% DMI ($p < 0.05$); however there was no significant difference in pH before feeding. The supplementation of CRP did not affect temperature and ammonia–nitrogen in the rumen, and total VFA, acetate ($C_2$), and butyrate ($C_4$) concentrations were similar among treatments. However, propionate ($C_3$) concentration before feeding and the average post-feeding values were significantly increased in the CRP group at 2% DMI ($p < 0.05$) compared to other groups, though there were no differences between the control and 6% DMI CRP groups. In addition, the $C_2/C_3$ ratio was reduced by the CRP supplementation at 2% DMI group ($p < 0.05$). Moreover, $CH_4$ production was significantly reduced following the 2% DMI CRP supplementation group, though was similar between the control and 4 and 6% DMI CRP groups. The results are presented in Table 3.

**Table 3.** Effect of chaya leaf and rambutan peel pellet (CRP) on rumen fermentation.

| Item | CRP (% of Dry Matter Intake) | | | | SEM | Contrast | | |
|---|---|---|---|---|---|---|---|---|
| | 0 | 2 | 4 | 6 | | L | Q | C |
| **Rumen Fermentation Characteristic** | | | | | | | | |
| pH | | | | | | | | |
| H0 | 7.0 | 7.1 | 6.9 | 6.7 | 0.049 | 0.058 | 0.383 | 0.435 |
| H4 | 6.7 [a] | 6.8 [a] | 6.5 [b] | 6.5 [b] | 0.019 | 0.001 | 0.136 | 0.043 |
| mean | 6.9 [a] | 6.9 [a] | 6.7 [ab] | 6.6 [b] | 0.031 | 0.011 | 0.251 | 0.198 |
| Temperature (°C) | | | | | | | | |
| H0 | 37.0 | 36.5 | 37.6 | 37.8 | 0.461 | 0.133 | 0.897 | 0.567 |
| H4 | 37.9 | 37.9 | 37.1 | 37.2 | 0.304 | 0.320 | 0.994 | 0.564 |
| mean | 37.5 | 37.2 | 37.4 | 37.5 | 0.129 | 0.148 | 0.109 | 0.104 |
| $NH_3N$ (mg/dL) | | | | | | | | |
| H0 | 16.0 | 16.5 | 14.6 | 14.7 | 1.021 | 0.540 | 0.931 | 0.659 |
| H4 | 26.3 | 24.4 | 26.0 | 24.7 | 0.700 | 0.619 | 0.823 | 0.350 |
| mean | 21.2 | 20.4 | 20.2 | 19.7 | 0.727 | 0.505 | 0.963 | 0.876 |
| Acetic acid (%) | | | | | | | | |
| H0 | 66.5 | 63.1 | 64.9 | 65.9 | 0.504 | 0.201 | 0.588 | 0.083 |
| H4 | 64.8 | 62.3 | 64.7 | 64.4 | 0.506 | 0.764 | 0.357 | 0.126 |
| mean | 65.7 | 62.7 | 64.8 | 65.1 | 0.430 | 0.344 | 0.392 | 0.064 |
| Propionic acid (%) | | | | | | | | |
| H0 | 22.1 [c] | 26.2 [a] | 24.3 [b] | 23.1 [bc] | 0.428 | 0.096 | 0.586 | 0.027 |
| H4 | 25.2 | 27.1 | 25.5 | 24.7 | 0.378 | 0.193 | 0.169 | 0.218 |
| mean | 23.7 [b] | 26.7 [a] | 24.9 [ab] | 23.8 [b] | 0.323 | 0.074 | 0.243 | 0.034 |
| Butyric acid (%) | | | | | | | | |
| H0 | 11.3 | 10.7 | 10.9 | 11.0 | 0.437 | 0.798 | 0.927 | 0.671 |
| H4 | 10.0 | 10.6 | 9.8 | 11.2 | 0.165 | 0.055 | 0.617 | 0.062 |
| mean | 10.7 | 10.7 | 10.3 | 11.1 | 0.480 | 0.376 | 0.935 | 0.743 |
| Total VFA | | | | | | | | |
| H0 | 94.1 | 97.8 | 104.8 | 99.1 | 1.955 | 0.261 | 0.275 | 0.401 |
| H4 | 98.6 | 96.1 | 102.3 | 94.2 | 1.278 | 0.555 | 0.315 | 0.089 |
| mean | 96.4 | 97.0 | 103.5 | 96.6 | 1.086 | 0.482 | 0.135 | 0.092 |
| $C_2/C_3$ Ratio | | | | | | | | |
| H0 | 3.4 | 2.5 | 2.8 | 3.1 | 0.112 | 0.140 | 0.933 | 0.059 |
| H4 | 2.6 | 2.3 | 2.6 | 2.6 | 0.056 | 0.398 | 0.222 | 0.186 |
| mean | 3.0 [a] | 2.4 [d] | 2.7 [c] | 2.9 [b] | 0.060 | 0.092 | 0.496 | 0.029 |
| $CH_4$ mM/L * | | | | | | | | |
| H0 | 28.4 [a] | 25.5 [c] | 26.9 [b] | 27.7 [ab] | 0.304 | 0.094 | 0.573 | 0.026 |
| H4 | 26.2 | 24.8 | 26.1 | 26.7 | 0.280 | 0.222 | 0.180 | 0.207 |
| mean | 27.3 [a] | 25.1 [b] | 26.5 [ab] | 27.2 [a] | 0.236 | 0.081 | 0.246 | 0.034 |

[a–c] Means in the same row with different superscripts significantly differ ($p < 0.05$); CRP = chaya powder and rambutan peel powder pellet, * $CH_4$ estimation by Moss et al. (2000); $CH_4$ production = 0.45 (acetate) − 0.275 (propionate) + 0.4 (butyrate); SEM = standard error of the mean.

### 3.3. Feed Intake and Nutrients Digestibility

Table 4 shows that in terms of %BW, rice straw intake was significantly higher in the CRP groups at 2 and 4% DMI but reduced in the 6% DMI CRP group compared to other groups ($p < 0.05$). There was no change of feed rice straw intake in term of kg/day. The concentrate intake was similar in all treatments, and CRP intake was linearly increased in the supplementation groups ($p < 0.05$). Furthermore, total feed intake in terms of kg/day was higher in the 2 and 6% DMI CRP groups, and the 4% DMI CRP group demonstrated an improved total DM intake in terms of %BW ($p < 0.05$) compared to other groups. Nevertheless, the apparent digestibilities of DM, OM, CP, NDF, NDF, and EE were similar among treatments.

**Table 4.** Effect of chaya leaf and rambutan peel pellet (CRP) a on feed intake and digestibility.

| Items | CRP (% of Dry Matter Intake) | | | | SEM | Contrast | | |
|---|---|---|---|---|---|---|---|---|
| | 0 | 2 | 4 | 6 | | L | Q | C |
| Rice straw intake | | | | | | | | |
| kg/day | 3.5 | 3.6 | 3.4 | 3.5 | 0.032 | 0.644 | 0.937 | 0.111 |
| %BW | 1.5 [c] | 1.6 [b] | 1.7 [a] | 1.3 [d] | 0.016 | 0.022 | 0.001 | 0.034 |
| Concentrate intake | | | | | | | | |
| kg/day | 2.0 | 2.0 | 2.0 | 2.0 | 0.042 | 0.421 | 0.264 | 0.625 |
| %BW | 0.8 | 0.9 | 1.0 | 0.9 | 0.065 | 0.124 | 0.245 | 0.641 |
| CRP intake | | | | | | | | |
| kg/day | 0.0 [d] | 0.1 [c] | 0.2 [b] | 0.3 [a] | 0.001 | <0.001 | 0.145 | 0.235 |
| %BW | 0.0 [b] | 0.1 [a] | 0.1 [a] | 0.1 [a] | 0.001 | <0.001 | 0.112 | 0.241 |
| Total DM intake | | | | | | | | |
| kg/day | 5.5 [b] | 5.7 [a] | 5.6 [ab] | 5.8 [a] | 0.032 | 0.022 | 0.937 | 0.111 |
| %BW | 2.3 [c] | 2.6 [b] | 2.8 [a] | 2.3 [c] | 0.018 | 0.147 | 0.010 | 0.001 |
| Apparent digestibility, % | | | | | | | | |
| Dry matter | 61.7 | 63.0 | 60.2 | 62.1 | 0.894 | 0.826 | 0.881 | 0.323 |
| Organic matter | 65.4 | 66.6 | 64.0 | 65.4 | 0.872 | 0.760 | 0.969 | 0.351 |
| Crude protein | 55.5 | 60.3 | 60.8 | 60.4 | 1.061 | 0.145 | 0.255 | 0.707 |
| Neutral detergent fiber | 45.0 | 47.3 | 42.7 | 47.1 | 1.079 | 0.879 | 0.654 | 0.149 |
| Acid detergent fiber | 36.8 | 35.8 | 29.5 | 29.3 | 1.750 | 0.119 | 0.922 | 0.490 |
| Ether extract | 18.2 | 82.0 | 82.0 | 82.9 | 1.131 | 0.217 | 0.553 | 0.661 |

[a–c] Means in the same row with different superscripts significantly differ ($p < 0.05$); CRP = chaya powder and rambutan peel powder pellet; SEM = standard error of the mean.

### 3.4. Nitrogen Balance

The effect of CRP supplementation on nitrogen balance is shown in Table 5. Increasing levels of CRP linearly increased nitrogen intake ($p < 0.05$). The nitrogen excretion of urine was reduced when cattle were fed CRP at 2% DMI and was significantly increased at in the 4 and 6% DMI CRP groups ($p < 0.05$). However, feces and total urine excretion were similar among treatments. Furthermore, the nitrogen absorption (g/d) and nitrogen retention (g/d) were significantly increased in the CRP supplementation groups ($p < 0.05$). Moreover, nitrogen absorption (% of nitrogen intake) and nitrogen retention (% of nitrogen intake) were higher in the 2 and 4% DMI CRP groups ($p < 0.05$), though there were no changes at 6% DMI.

**Table 5.** Effect of chaya leaf and rambutan peel pellet (CRP) on nitrogen balance.

| Items | CRP (% of Dry Matter Intake) | | | | SEM | Contrast | | |
|---|---|---|---|---|---|---|---|---|
| | 0 | 2 | 4 | 6 | | L | Q | C |
| Nitrogen intake, g/d | 68.0 [d] | 71.7 [c] | 74.1 [b] | 77.7 [a] | 0.150 | 0.037 | 0.111 | 0.111 |
| Nitrogen excretion, g/d | | | | | | | | |
| Feces | 24.1 | 22.9 | 22.3 | 26.8 | 1.235 | 0.549 | 0.278 | 0.709 |
| Urine | 8.8 [b] | 7.6 [c] | 9.7 [a] | 9.6 [a] | 0.201 | 0.048 | 0.020 | 0.222 |
| Total | 32.9 | 30.5 | 32.0 | 36.4 | 1.217 | 0.333 | 0.198 | 0.915 |
| Nitrogen absorption, g/d | 43.8 [b] | 48.8 [a] | 51.8 [a] | 50.9 [a] | 0.234 | 0.029 | 0.274 | 0.877 |
| Nitrogen retention, g/d | 34.9 [b] | 41.2 [a] | 42.1 [a] | 41.3 [a] | 0.203 | 0.018 | 0.195 | 0.743 |
| Nitrogen absorption, % of nitrogen intake | 64.3 [b] | 68.1 [a] | 69.9 [a] | 65.5 [b] | 0.671 | 0.037 | 0.267 | 0.086 |
| Nitrogen retention, % of nitrogen intake | 51.3 [b] | 57.5 [a] | 56.8 [a] | 53.1 [ab] | 0.636 | 0.045 | 0.183 | 0.080 |

[a–c] Means in the same row with different superscripts significantly differ ($p < 0.05$); CRP = chaya powder and rambutan peel powder pellet; SEM = standard error of the mean.

## 4. Discussions

Generally, animal feeds from fodder or shrubs have been shown to contain high levels of protein (20–30% CP), carbohydrate, and phytonutrients [2,20], and the result of this experiment were similar, since the CRP pellets comprised 20.1% CP, 5.1% CT, and 4.8% SP. The CRP supplement also contained essential amino acids and macro minerals. These phytonutrients, especially SP, could inhibit protozoa and methanogen levels, and CT could affect some cellulolytic bacteria and adhering to the surface of feed particles [11].

In this experiment, rumen pH at 4 h and after feeding was slightly reduced from 6.9 to 6.6, which was in the normal range when the animals received concentrate and roughage sources. Avila et al. [21] similarly found that the supplementation of CT at 20 g/kg of diet DM linearly reduced rumen pH in Jersey steers. Likewise, Bodas et al. [22] reported that the rumen pH tended to be high before feeding and reached a low level between 3 and 6 h post-feeding. A ruminal pH of 6.5–6.8 was reported to be an optimal range to improve rumen fermentation, microbial protein synthesis, and production in ruminants [17]. Ruminal temperature generally ranges from 37 to 39 °C, which is a suitable level for rumen ecology and microbial enzyme activity [23], and the values found in this experiment (37–38 °C) were similar. $NH_3$-N concentrations were not different among treatments. Preston and Leng [24] and Wanapat and Pimpa [25] stated that ruminal $NH_3$-N levels within 15–30 mg/dL significantly enhanced rumen fermentation and were suitable for microbial activity, as also shown in the present study (14–26 mg/dL).

Total VFA, $C_2$, and $C_4$ levels were similar among treatments; however, $C_3$ was significantly increased by the supplementation of CRP at 2% DMI, which agreed with the results of Totakul et al. [6], who found that the supplementation of chaya leaf pellets in crossbred bulls did not influence $C_2$ and $C_4$ concentrations but did enhance total VFA and $C_3$ concentration. The supplementation of fruit peels containing PTN was found to improve total VFA and $C_3$, and no changes of $C_2$ and $C_4$ concentrations were reported [26]. In addition, Cherdthong et al. [11] revealed that rechanneling hydrogen in the methane production pathway led to enhanced $C_3$ synthesis. Moreover, Avila et al. [21] found that CT extract supplementation significantly reduced the $C_2/C_3$ ratio, which matches the result of this study, in which CRP supplementation at 2, 4 and 6% DMI remarkably reduced the $C_2/C_3$ ratio and CRP supplementation at 2% DMI deceased methane production. In feed, CT and SP are able to mitigate methane production and manipulate volatile fatty acid proportions in ruminants. As mentioned earlier, PTN influences protozoal populations and methanogen activities, resulting in reduced methane production [27]. Montoya-Flores et al. [28] found that using dried leaves of *Leucaena leucocephala* at 12% DMI could reduce $CH_4$ production in crossbred heifers. Patra and Saxana [29] showed that CT reduced protozoa populations via defaunation and blocked the activity of methanogens, which could reduce methane, increase microbial growth, and increase the level of bypass proteins in the lower gut. Hassanat and Benchaar [30] also found that using condensed tannins and hydrolysable

tannins in vitro could reduce methane production with no adverse effects on fermentation end-products. In addition, Ampapon and Wanapat [31] found that the supplementation of rambutan peel powder in dairy cows reduced the protozoal population and mitigated methane production. Recently, Totakul [5] reported that chaya leaf pellets improved rumen fermentation and mitigated methane production via an in vitro gas production technique. Furthermore, Phesatcha et al. [32] found that the use of *Flemingia macrophylla* with PTN, as protein source fodder, could mitigate methane production in beef cattle.

In this study, rice straw intake and total DM intake were increased in the groups supplemented with CRP at 2 and 4% DMI, but there was no change at 6% DMI; the nutrient digestibilities were similar. These results agree with those of Totakul et al. [6], who also found that using chaya leaf pellets increased total DMI, though digestibility levels were again similar. The supplementation of topical legumes such as *Leucaena* leaf can improve total DMI and digestibility [33]. Archimède et al. [34] stated that tropical tannin-rich plants such as *G. sepium*, *L. leucocephala*, and *M. esculenta* in Blackbelly sheep improved DMI without additional effects on nutrients digestibility. Recently, Phesatcha et al. [32] reported that using *Flemingia* as protein source fodder did not affect DMI but did improve the apparent nutrient digestibility. Similarly, Montoya-Flores et al. [28] reported that using dried leaves of *Leucaena leucocephala* did not affect DMI in crossbred heifers. In addition, pelleted hazel leaf supplementation reduced nutrient digestibilities in sheep, possibly due to high concentrations of tannins [35]. Likewise, Arisya et al. [12] showed that the supplementation of tannin extracts in an in vitro study reduced DM digestibility and protein degradation. On the other hand, Ampapon et al. [36] stated that the use of dietary sources (mangosteen peel and banana flower powder) containing phytonutrients fed at 100 g/day did not negatively affect DMI and nutrient digestibility; similar results were reported by Ampapon and Wanapat [9], who found that using rambutan peel powder did not change feed intake and DM digestibility.

In this experiment, nitrogen intake, nitrogen absorption and nitrogen retention were significantly increased after supplementation with CRP at 2 and 4% DMI, though nitrogen excretion was similar across groups. Similarly, Archimède et al. [34] showed that tropical tannin-rich plants such as *G. sepium*, *L. leucocephala*, and *M. esculenta* in B. *Blackbelly* sheep increased nitrogen intake, absorption, and retention while reducing urine nitrogen excretion. Ampapon et al. [23] reported that the supplementation of cassava hay at 400 g/day improved nitrogen intake, absorption, and retention in buffaloes. In addition, the use of topical legumes such as *Leucaena* leaves was shown to increase nitrogen intake, excretion, absorption, and retention in buffaloes [33]. Furthermore, royal poinciana seed meal pellet supplementation at 100 g/day in Thai native beef cattle was shown to enhance nitrogen utilization and microbial protein synthesis [11]. The discrepancies observed in a number of experiments could be attributed to the phytonutrients used in each.

## 5. Conclusions

In this experiment, the supplementation of CRP at 2 and 4% DMI enhanced rumen fermentation, specifically by improving $C_3$ concentration, reducing the C2/C3 ratio, mitigating methane production, and enhancing feed intake and nitrogen utilization efficiency. Therefore, CRP could be used as a rumen enhancer.

**Author Contributions:** T.A.: planned and conducted the feed experiment, conducted samplings, drafted the manuscript, and conducted the chemical analyses and tabulation.; M.W. supervised the design and execution of the experiment, interpreted data, commented on and supervised the writing of the manuscript, and submitted the paper.; T.H.: statistically analyzed the interaction of data and commented on and supervised the writing of manuscript. All authors have read and agreed to the published version of the manuscript.

**Funding:** This work was supported by Post-Doctoral Training Scholarship (PD2562-01), Khon Kaen University, Thailand and Fundamental Fund (FF) project (no. 65A103000130), Khon Kaen University, Thailand and Fundamental Fund (FF) project (no. FRB650059/SRN/01-4), Rajamangala University of Technology Isan Surin Campus, Surin 32000, Thailand, under administration of MHESI, Thailand.

**Institutional Review Board Statement:** The study procedure conducted under approval procedure by the Animal Care and Use Committee of Khon Kaen University and carried out by the Institute of Animals for Scientific Purpose Development (IAD), Thailand (record no. U1-06878-2560).

**Informed Consent Statement:** Not applicable.

**Data Availability Statement:** Not applicable.

**Acknowledgments:** The authors would like to thank the Ministry of Higher Education, Science, Research and Innovation (MHESI), Thailand; Tropical Feed Resources Research and Development Center (TROFREC), Department of Animal Science, Faculty of Agriculture, Khon Kaen University, Thailand; and Department of Animal Science, Faculty of Agriculture and Technology, Rajamangala University of Technology Isan Surin Campus, Surin 32000, Thailand for providing research facilities and financial support.

**Conflicts of Interest:** The authors declare no conflict of interest.

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
