# Peer review of "Enhancing Rumen Fermentation Characteristic and Methane Mitigation Using Phytonutrient Pellet in Beef Cattle"

_fermentation, doi:10.3390/fermentation8050239_

Round 1

Reviewer 1 Report

Previous study showed that chaya has a good nutritional profile and can be a potential source of quality feed for livestock. One study showed that all the amino acids required to support 10 kg/d of milk production were met by feeding a diet composed solely of chaya fodder. Also, authors of this manuscript showed that increasing supplementation level of CHYP linearly enhanced (p < 0.05) DM, OM, and CP digestibilities. The manuscript is very well written. There are no methodological omissions or inconsistencies in the work. Decrease in CH4 production and increase nitrogen utilization efficiency is interesting result. The manuscript should be accepted.

Author Response

Dear Editor and Reviewers,

We again wish to express our most sincere thanks for your kind support and for valuable suggestions and recommendations. We have carefully revised it and have it corrected as suggested.

            Thanks a lot for the prompt support.

Thanks for your support

With warm regards,

Prof. Dr. Metha Wanapat

Contact: [email protected]

The main corrections in the paper and the responds to the reviewer’s comments are as flowing:

Reviewer 1

Previous study showed that chaya has a good nutritional profile and can be a potential source of quality feed for livestock. One study showed that all the amino acids required to support 10 kg/d of milk production were met by feeding a diet composed solely of chaya fodder. Also, authors of this manuscript showed that increasing supplementation level of CHYP linearly enhanced (p < 0.05) DM, OM, and CP digestibilities. The manuscript is very well written. There are no methodological omissions or inconsistencies in the work. Decrease in CH4 production and increase nitrogen utilization efficiency is interesting result. The manuscript should be accepted.

Response: Thanks for the constructive comments and we fully agreed. Under this experiment the combinational effect of level of essential amino acids and phytonutrients especially from Chaya leaf and rambutan peel powder have significantly enhanced rumen fermentation and feed utilization.

Reviewer 2 Report

Manuscript fermentation-1685416, entitled “Enhancing rumen fermentation characteristic and methane mitigation using phytonutrient pellet in beef cattle”

Recommendation:       The above paper is not suitable for publication in its present form.

General Comments:

  • This article is intended to provide information about the effects of a phytonutrient pellet on rumen fermentation characteristic and methane mitigation in beef cattle. My main concern is the too small sample size. One animal per treatment and period.
  • You prepare CRP pellet. How was it then mixed with the concentrate? You have mixed the concentrate pellets with that of CRP at the denoted levels? How was sorting (by the animals) avoided?
  • L89: Furthermore, 2 kg contained CRP at 2, 4 and 6% or CRP pellets were additionally provided?
  • Please provide a uniform presentation; ration of C2/C3 or A:P ratio
  • Please clarify the use of superscripts (for example, Table 5, Urine: what is the meaning of “b,b”, “c,c” etc? The same in Table 3, pH – H4)
  • L149-150: Was this finding not expected? Toy provided 3 groups with 2, 4 and 6% CRP.
  • In some parts authors reach to incorrect conclusions:
  • L135: Please check superscripts. When the same superscript is shown, no significant differences are denoted (not for 4%)
  • L150-151: Only in 2% and 6% CRP group. According to kg/day in 2% and according to %BW in 4%.
  • L192, 200: 4%? Please check Table 3. Similar superscripts
  • L227-236: How are these discrepancies explained?

Specific Comments:

L19: “…ature, ammonia nitrogen and total volatile fatty acids…”

L25: “by the treatment” instead of “among treatments”

L34: “contributor” instead of “contributions”

L35-36: “There is a number of factors that could enhance rumen fermentation, such as the sources of protein and carbohydrate during various seasons [1].”

L42: “has been” instead of “was”

L43-44: “…protein (29-31%), essential amino acids, and minerals, as well as flavonoids, tannins…”

L45-46: “…vivo studies can improve nutrients’ digestibility, rumen fermentation and production efficiency in cattle [5,6].”

L47-48: “…receiving a great concern from all sectors and enteric fermentation has contributed to this phenomenon by rumen methane…”

L49: “emission” instead of “pollution”

L57: “alleviate” instead of “mitigated”

L61: “with” instead of “to contain more”

L62: Please delete “respectively”

L65: “…the effects of supplementation with Chaya…”

L66: “intake” instead of “consumption”

L70: What do you mean by “from the planted with 60 days of regrowth”?

L74-76: “…screen mill. CRP pellet consisted of chaya leaf power (70%), rambutan peel powder (20%), cassava powder (9%), and 75 molasses (1%). All the…”

L87-88: “…(T2, T3, and T4, respectively) and feed composition is reported in Table 2.”

L93-94: “Feed samples were collected during the first two weeks, while feed intake at dry matter basis was daily measured.”

L96: “collected” instead of “taken”

L99-101: “…were filtered and 45 ml were collected and mixed with 5 ml of H2SO4 solution (1 M) to stop microbial activity for further volatile fatty acids (VFA) and ammonia nitrogen (NH3-N) analyses. The rumen…”

L104: “On the same day of rumen fluid…”

L106: “centrifuged” instead of “centrifugation”

L107-108: “…analysis as already described by Crocker [13]. Feeds…”

L111: “as described by” instead of “followed the report of”

L122: Please delete “respectively”

L124: “but also” instead of “while CRP contained”

L125: Please delete “amino acid profile of”

L126: “…alanine and other amino acids at the levels shown in Table 1.”

L132: “At the same time” instead of “While the”

L159: Please delete “as”

L173-174: “…in this experiment, since CRP pellets contained 20.1% CP, 5.1% CT and 4.8% SP. In addition…”

L174-175: “…CRP contained essential amino acids and macro…”

L176-177: Please rephrase

L187-188: “different among” instead of “changed in all”

L189: “within” instead of “from”

L190: “…for microbial activity, as shown in the present study (14-26 mg/dL).”

L194: Please delete “on”

L199: “as shown in the present” instead of “and was similar to this”

L209-210: “…tannins in vitro can reduce…”

L214: “by an” instead of “in”

L223: Please delete “and”

L225, 227, 233: Please delete “on”

L230: “…in an in vitro study…”

L233-234: “as similarly indicated by” instead of “and was similar to the report of”

L234: “found” instead of “stated”

L235: This?

L238-239: “…increased after supplementation with CRP at 2 and 4% of DMI. Nevertheless…”

L247-249: “The discrepancies observed in a number of experiments could be attributed to phytonutrients…”

L255: “could” instead of “should”

Author Response

Dear Editor and Reviewers,

We again wish to express our most sincere thanks for your kind support and for valuable suggestions and recommendations. We have carefully revised it and have it corrected as suggested.

            Thanks a lot for the prompt support.

Thanks for your support

With warm regards,

(Prof. Dr. Metha Wanapat)

Contact: [email protected]

The main corrections in the paper and the responds to the reviewer’s comments are as flowing:

Reviewer 2

General Comments:

This article is intended to provide information about the effects of a phytonutrient pellet on rumen fermentation characteristic and methane mitigation in beef cattle. My main concern is the too small sample size. One animal per treatment and period.

Response: Thanks for the useful comments and we agreed. Nevertheless, standard design of  4×4 Latin square design has been imposed continuously to study the feed digestibility and rumen metabolism in ruminants before further in vivo feeding trial can be investigated.

You prepare CRP pellet. How was it then mixed with the concentrate? You have mixed the concentrate pellets with that of CRP at the denoted levels? How was sorting (by the animals) avoided?

Response: Thanks, we have added as show in the text.

L89: Furthermore, 2 kg contained CRP at 2, 4 and 6% or CRP pellets were additionally provided?

Response: Thanks, we have revised as show in the text.

Please provide a uniform presentation; ration of C2/C3 or A:P ratio

Response: Thanks, we have added as show in the text.

Please clarify the use of superscripts (for example, Table 5, Urine: what is the meaning of “b,b”, “c,c” etc? The same in Table 3, pH – H4)

Response: Thanks, we have added already, as show in the text.

L149-150: Was this finding not expected? Toy provided 3 groups with 2, 4 and 6% CRP.

Response: Thanks, we have revised and added as show in the text.

In some parts authors reach to incorrect conclusions:

L135: Please check superscripts. When the same superscript is shown, no significant differences are denoted (not for 4%)

Response: Thanks, we have added as show in the text.

L150-151: Only in 2% and 6% CRP group. According to kg/day in 2% and according to %BW in 4%.

Response: Thanks, we have added as show in the text.

L192, 200: 4%? Please check Table 3. Similar superscripts

Response: Thanks, we have added as show in the text.

L227-236: How are these discrepancies explained?

Response: Thanks, we have added as show in the text.

Specific Comments:

L19: “…ature, ammonia nitrogen and total volatile fatty acids…”

Response: Thanks, we have revised already.

L25: “by the treatment” instead of “among treatments”

Response: Thanks, we have revised already.

L34: “contributor” instead of “contributions”

Response: Thanks, we have revised already.

L35-36: “There is a number of factors that could enhance rumen fermentation, such as the sources of protein and carbohydrate during various seasons [1].”

Response: Thanks, we have revised already.

L42: “has been” instead of “was”

Response: Thanks, we have revised already.

L43-44: “…protein (29-31%), essential amino acids, and minerals, as well as flavonoids, tannins…”

Response: Thanks, we have revised already.

L45-46: “…vivo studies can improve nutrients’ digestibility, rumen fermentation and production efficiency in cattle [5,6].”

Response: Thanks, we have revised already.

L47-48: “…receiving a great concern from all sectors and enteric fermentation has contributed to this phenomenon by rumen methane…”

Response: Thanks, we have revised already.

L49: “emission” instead of “pollution”

Response: Thanks, we have revised already.

L57: “alleviate” instead of “mitigated”

Response: Thanks, we have revised already.

L61: “with” instead of “to contain more”

Response: Thanks, we have revised already.

L62: Please delete “respectively”

Response: Thanks, we have revised already.

L65: “…the effects of supplementation with Chaya…”

Response: Thanks, we have revised already.

L66: “intake” instead of “consumption”

Response: Thanks, we have revised already.

L70: What do you mean by “from the planted with 60 days of regrowth”?

Response: Thanks, we have revised already.

L74-76: “…screen mill. CRP pellet consisted of chaya leaf power (70%), rambutan peel powder (20%), cassava powder (9%), and 75 molasses (1%). All the…”

Response: Thanks, we have revised already.

L87-88: “…(T2, T3, and T4, respectively) and feed composition is reported in Table 2.”

Response: Thanks, we have revised already.

L93-94: “Feed samples were collected during the first two weeks, while feed intake at dry matter basis was daily measured.”

Response: Thanks, we have revised already.

L96: “collected” instead of “taken”

Response: Thanks, we have revised already.

L99-101: “…were filtered and 45 ml were collected and mixed with 5 ml of H2SO4 solution (1 M) to stop microbial activity for further volatile fatty acids (VFA) and ammonia nitrogen (NH3-N) analyses. The rumen…”

Response: Thanks, we have revised already.

L104: “On the same day of rumen fluid…”

Response: Thanks, we have revised already.

L106: “centrifuged” instead of “centrifugation”

Response: Thanks, we have revised already.

L107-108: “…analysis as already described by Crocker [13]. Feeds…”

Response: Thanks, we have revised already.

L111: “as described by” instead of “followed the report of”

Response: Thanks, we have revised already.

L122: Please delete “respectively”

Response: Thanks, we have revised already.

L124: “but also” instead of “while CRP contained”

Response: Thanks, we have revised already.

L125: Please delete “amino acid profile of”

Response: Thanks, we have revised already.

L126: “…alanine and other amino acids at the levels shown in Table 1.”

Response: Thanks, we have revised already.

L132: “At the same time” instead of “While the”

Response: Thanks, we have revised already.

L159: Please delete “as”

Response: Thanks, we have revised already.

L173-174: “…in this experiment, since CRP pellets contained 20.1% CP, 5.1% CT and 4.8% SP. In addition…”

Response: Thanks, we have revised already.

L174-175: “…CRP contained essential amino acids and macro…”

Response: Thanks, we have revised already.

L176-177: Please rephrase

Response: Thanks, we have revised already.

L187-188: “different among” instead of “changed in all”

Response: Thanks, we have revised already.

L189: “within” instead of “from”

Response: Thanks, we have revised already.

L190: “…for microbial activity, as shown in the present study (14-26 mg/dL).”

Response: Thanks, we have revised already.

L194: Please delete “on”

Response: Thanks, we have revised already.

L199: “as shown in the present” instead of “and was similar to this”

Response: Thanks, we have revised already.

L209-210: “…tannins in vitro can reduce…”

Response: Thanks, we have revised already.

L214: “by an” instead of “in”

Response: Thanks, we have revised already.

L223: Please delete “and”

Response: Thanks, we have revised already.

L225, 227, 233: Please delete “on”

Response: Thanks, we have revised already.

L230: “…in an in vitro study…”

Response: Thanks, we have revised already.

L233-234: “as similarly indicated by” instead of “and was similar to the report of”

Response: Thanks, we have revised already.

L234: “found” instead of “stated”

Response: Thanks, we have revised already.

L235: This?

Response: Thanks, we have deleted already.

L238-239: “…increased after supplementation with CRP at 2 and 4% of DMI. Nevertheless…”

Response: Thanks, we have revised already.

L247-249: “The discrepancies observed in a number of experiments could be attributed to phytonutrients…”

Response: Thanks, we have revised already.

L255: “could” instead of “should”

Response: Thanks, we have revised already.

Reviewer 3 Report

The manuscript “Enhancing rumen fermentation characteristic and methane mitigation using phytonutrient pellet in beef cattle” is relevant for the Fermentation journal and presented in a well-structured manner. The manuscript scientifically sounds and has a high applied value. The manuscript’s results seem to be reproducible based on the details given in the methods section. Some necessary minor corrections I have indicated below. The tables are appropriate, and reflect the results properly. The data are interpreted appropriately and consistently throughout the manuscript. The statistical analysis is adequate to the data obtained and conclusions. The conclusions are consistent with the evidence and arguments presented.

The following inconsistencies should be corrected.

  1. Although, the analysis of the rumen microbial communities was not an object of this study, everyone can recognize that all characteristics of the rumen fluid as well as the methane production are determined mainly by the rumen microbiome. Therefore, some examples of the rumen microbiome changes influenced by phytonutrients or their substances are very desirable in the Discussion chapter. Moreover, some suggestions on the changes in the rumen microbial composition triggered by the Chaya leaf and rambutan and interrelated with known facts from literature in this area, would be improve understanding of the physiological and biochemical findings described in the article.
  2. Lines 24-25. The following abbreviations in the abstract should be replaced by full words “DM, OM, CP, NDF, NDF, and EE”. NDF is mentioned twice due to misprint, isn’t it?
  3. Line 110. “The protocols” should be replaced by “The protocol”.
  4. Line 121-124. All abbreviations in the paper should be explained at the first appearance, such as “OM, CP, TDN, NDF, ADF, etc.”.
  5. Line 153. All abbreviations in the paper should be explained at the first appearance, such as “DM, OM, CP, NDF, NDF, and EE”. NDF is mentioned twice due to misprint, isn’t it?
  6. Line 159. “is as shown” should be replaced by “is shown”.
  7. Line 185. “Ranges” should be replaced by “ranged”.
  8. Line 197. “That” should be removed.
  9. Line 198. “Was found” should be removed.
  10. Line 201. “Indicators” is not exact and comprehensive term for description of the effects of tannins and saponins to the methane production. Perhaps, the term “agents” or “chemical agents” would be better.
  11. Line 218. Dot after “change” should be replaced by comma.

Author Response

Dear Editor and Reviewers,

We again wish to express our most sincere thanks for your kind support and for valuable suggestions and recommendations. We have carefully revised it and have it corrected as suggested.

            Thanks a lot for the prompt support.

Thanks for your support

With warm regards,

(Prof. Dr. Metha Wanapat)

Contact: [email protected]

The main corrections in the paper and the responds to the reviewer’s comments are as flowing:

Reviewer 3

The manuscript “Enhancing rumen fermentation characteristic and methane mitigation using phytonutrient pellet in beef cattle” is relevant for the Fermentation journal and presented in a well-structured manner. The manuscript scientifically sounds and has a high applied value. The manuscript’s results seem to be reproducible based on the details given in the methods section. Some necessary minor corrections I have indicated below. The tables are appropriate, and reflect the results properly. The data are interpreted appropriately and consistently throughout the manuscript. The statistical analysis is adequate to the data obtained and conclusions. The conclusions are consistent with the evidence and arguments presented.

The following inconsistencies should be corrected.

  1. Although, the analysis of the rumen microbial communities was not an object of this study, everyone can recognize that all characteristics of the rumen fluid as well as the methane production are determined mainly by the rumen microbiome. Therefore, some examples of the rumen microbiome changes influenced by phytonutrients or their substances are very desirable in the Discussion chapter. Moreover, some suggestions on the changes in the rumen microbial composition triggered by the Chaya leaf and rambutan and interrelated with known facts from literature in this area, would be improve understanding of the physiological and biochemical findings described in the article.

Response: Yes, many thanks for the insightful comments and suggestion. We fully agreed and have modified to demonstrate the effect of phytonutrient. Please see in the text.

  1. Lines 24-25. The following abbreviations in the abstract should be replaced by full words “DM, OM, CP, NDF, NDF, and EE”. NDF is mentioned twice due to misprint, isn’t it?

Response: Response: Thanks, we have revised already.

  1. Line 110. “The protocols” should be replaced by “The protocol”.

Response: Response: Thanks, we have revised already.

  1. Line 121-124. All abbreviations in the paper should be explained at the first appearance, such as “OM, CP, TDN, NDF, ADF, etc.”.

Response: Response: Thanks, we have revised already.

  1. Line 153. All abbreviations in the paper should be explained at the first appearance, such as “DM, OM, CP, NDF, NDF, and EE”. NDF is mentioned twice due to misprint, isn’t it?

Response: Response: Thanks, we have revised already.

  1. Line 159. “is as shown” should be replaced by “is shown”.

Response: Response: Thanks, we have revised already.

  1. Line 185. “Ranges” should be replaced by “ranged”.

Response: Response: Thanks, we have revised already.

  1. Line 197. “That” should be removed.

Response: Response: Thanks, we have revised already.

  1. Line 198. “Was found” should be removed

Response: Response: Thanks, we have revised already.

  1. Line 201. “Indicators” is not exact and comprehensive term for description of the effects of tannins and saponins to the methane production. Perhaps, the term “agents” or “chemical agents” would be better.

Response: Response: Thanks, we have revised already.

  1. Line 218. Dot after “change” should be replaced by comma.

Response: Response: Thanks, we have revised already.

Round 2

Reviewer 2 Report

Authors made the majority of the necessary amendments, however some points should be further clarified:

Two points were not answered by the authors:

You prepare CRP pellet. How was it then mixed with the concentrate? You have mixed the concentrate pellets with that of CRP at the denoted levels? How was sorting (by the animals) avoided?

Furthermore, 2 kg contained CRP at 2, 4 and 6% or CRP pellets were additionally provided?

Minor points

L58: Please rephrase

L268-269: Not for methane. Only 2%

In Table 3 please correct A:P ratio

L338-339: Please rephrase

Author Response

Dear Editor and Reviewers,

We again wish to express our most sincere thanks for your kind support and for valuable suggestions and recommendations. We have carefully revised it and have it corrected as suggested.

            Thanks a lot for the prompt support.

Thanks for your concern and kind facilitation

With warm regards,

(Prof. Dr. Metha Wanapat)

Contact: [email protected]

The main corrections in the paper and the responds to the reviewer’s comments are as flowing:

Reviewer 2

Two points were not answered by the authors:

You prepare CRP pellet. How was it then mixed with the concentrate? You have mixed the concentrate pellets with that of CRP at the denoted levels? How was sorting (by the animals) avoided?

Response: The CRP pellet was offered by well-mixing with the concentrate and there was no sorting. Well consumed.

Furthermore, 2 kg contained CRP at 2, 4 and 6% or CRP pellets were additionally provided?

Response: Yes, with CRP pellet included, thanks.

Minor points

L58: Please rephrase

Response: we have done it, thanks.

L268-269: Not for methane. Only 2%

Response: we have corrected it as suggested, thanks.

In Table 3 please correct A:P ratio

Response: we have corrected it as suggested, thanks.

L338-339: Please rephrase

Response: we have done it, thanks.
